# Sustainable Wastewater Management to Reduce Freshwater Contamination and Water Depletion in Mexico

José de Anda [1,*] and Harvey Shear [2]

1   Center for Research and Assistance in Technology and Design of the State of Jalisco,
    Department of Environmental Technology, C.A. Avenida Normalistas 800,
    Guadalajara 44270, Jalisco, Mexico
2   Institute for Management and Innovation, University of Toronto Mississauga, 3359 Mississauga Road,
    Mississauga, ON L5L 1C6, Canada; harvey.shear@utoronto.ca
*   Correspondence: janda@ciatej.mx; Tel.: +52-33-3345-5200

**Abstract:** At present, most rivers, lakes, and reservoirs in Mexico have significant anthropogenic contamination. The lack of sanitation infrastructure, the increase in the number of nonoperational or abandoned sanitation facilities, limited enforcement of environmental regulations, and limited public policies for the reuse of treated wastewater all contribute to the contamination and water availability problem. The reasons for this are identified as (1) the high maintenance and operational costs in sanitation facilities (including electricity consumption); (2) poor planning and practices of wastewater management and reuse by municipalities; (3) national policies that do not favor the reuse of treated wastewater for agriculture, industry, and municipal services instead of using groundwater as at present; (4) failure to adopt a governance model at the three levels of government; and (5) transparency in the management of financial resources. Some measures to improve this situation include (a) transparent decision-making; (b) participation and accountability in budgeting and planning at the national, state, and municipal levels; and (c) planning for the reuse of treated wastewater to reduce groundwater extractions and to reduce discharges to surface waters from the beginning of every WWTP project.

**Keywords:** municipal wastewater treatment plants; wastewater treatment systems; wastewater reuse; balanced scorecard; developing countries; Mexico

## 1. Introduction

Apart from their value as water sources for water supply for urban areas and food production, freshwater lakes have always been important to human life, because they serve as a freshwater fishery, recreation sites, and avenues of transport. They also provide other benefits, such as wildlife preservation, the replenishment of groundwater, flood regulation, regulation of the local climate, and enhancement of the beauty of local landscapes [1].

The continuing increase in global population is raising the demand of freshwater supply. One important factor affecting freshwater availability is associated with socioeconomic development and climate change. Another factor is the general lack of sanitation and waste treatment facilities in high-population areas of developing countries [2]. The quality of surface water or groundwater at any point in a watershed reflects the combined effect of many physical, chemical, and biological processes that affect water as it moves along hydrologic pathways over, under, and through the land [2]. Impacts on the water quality of freshwater ecosystems substantially reduce the water availability of regions [3]. Therefore, an efficient sanitation of wastewaters is becoming an issue that must be considered, particularly in developing countries where there is a major lag in providing this strategic service for the population.

At the end of 2016, in Mexico, the national coverage in the treatment of municipal wastewater was about 54% [4]. However, of the total installed municipal WWTPs, in the

same year, about 22.47% were out of operation. Some of the reasons for this were identified as the dominance in the country of a wastewater treatment technology demanding high electricity consumption, maintenance and operation costs, poor planning practices of municipal wastewater management and reuse, the failure to adopt a governance model, and transparency in the management of financial resources at the municipal level [5]. As a result, several water bodies in the Mexican hydrographic basins have been reported to be in an advanced process of eutrophication, increase of toxic contaminants, depletion of water levels, loss of habitat and aesthetic values, and impairment with recreational activities, among others [6].

On the other hand, most Mexico suffers under high water stress, and many regions of the country are highly vulnerable to droughts because of climate change; the contamination of its freshwater bodies will accelerate in a few years the process of the loss of water availability in many regions of the country [7–9]. Baseline water stress measures the ratio of total water withdrawals to the available renewable surface and groundwater supplies. Water withdrawals include domestic, industrial, irrigation, and livestock consumptive and non-consumptive uses. The available renewable water supplies include the impact of upstream consumptive water users and large dams on the downstream water availability. Higher values indicate more competition among users [10]. Figure 1 shows the main regions in the country suffering under water stress.

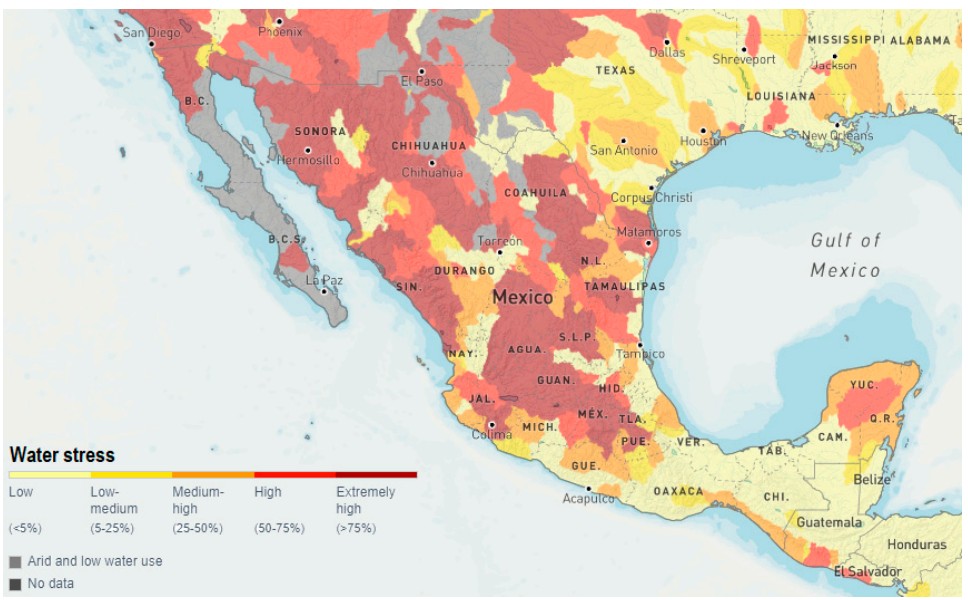

**Figure 1.** High-water stress in the central and northern areas of Mexico [10].

Baseline water depletion measures the ratio of total water consumption to the available renewable water supplies. The total water consumption includes domestic, industrial, irrigation, and livestock consumptive uses. The available renewable water supplies include the impact of upstream consumptive water users and large dams on the downstream water availability. Higher values indicate a larger impact on the local water supply and a decreased water availability for downstream users. Baseline water depletion is similar to baseline water stress; however, instead of looking at the total water withdrawal (consumptive plus non-consumptive), baseline water depletion is calculated using the consumptive withdrawal only [10]. Figure 2 shows the water depletion in Mexico affecting large central and northern regions of the territory.

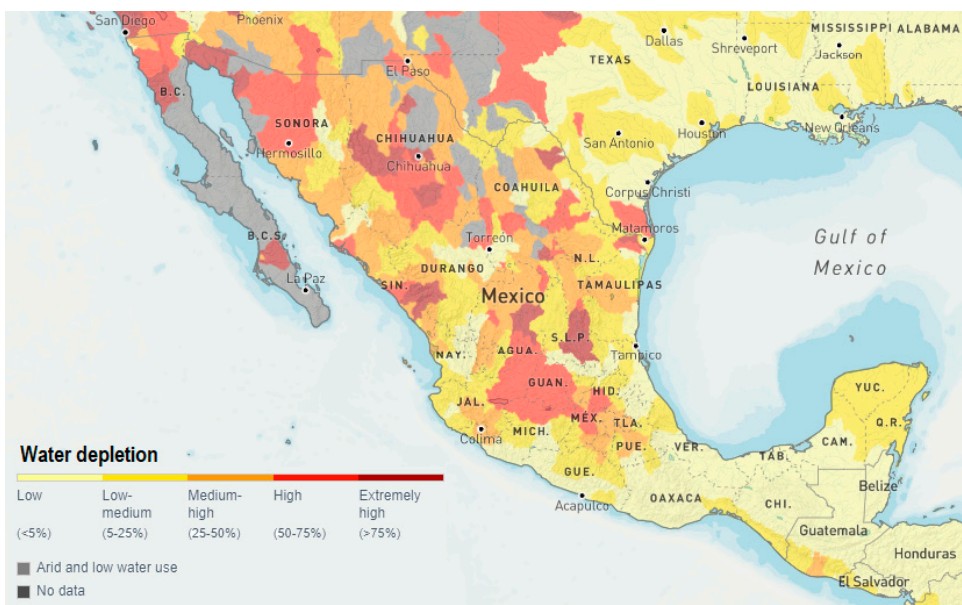

**Figure 2.** Water depletion in the central and northern areas of Mexico [10].

The purpose of present work is to discuss some of the reasons why, after years of investments in wastewater treatment projects, no substantial progress in sanitation coverage has been made in recent years. The current deficiencies in the national wastewater management model are significantly affecting all surface freshwater ecosystems in the country [4,7].

Some alternatives to improving wastewater management include: (1) strengthening the present water governance model at the three levels of government, (2) the adoption of integrated water resource management (IWRM) practices, and (3) the reuse of municipal treated wastewater for agriculture and the industry. These alternatives are proposed as potential strategies to advance the national commitment in relation to the sixth of the UN Sustainable Development Goals, SDG6: "Ensure access to water and sanitation for all" [11].

## 2. Materials and Methods

Historical data on the population in Mexico for the period 1950–2018 were obtained from the National Population Council (CONAPO, its acronym in Spanish) [12]. The projected growth of the population until 2050 was also included.

The National Water Commission (CONAGUA, its acronym in Spanish) is a decentralized administrative body of the Ministry of Environment and Natural Resources (SEMARNAT, its acronym in Spanish) created in 1989, whose responsibility is to administer, regulate, control, and protect the national waters in Mexico. Official data from CONAGUA and SEMARNAT were used to analyze the historical evolution of the wastewater and treatment infrastructure and coverage in Mexico for the period 1992–2017 [4,13]. Unfortunately, 2017 was the last date in which official wastewater and sanitation data were published by the federal agencies in Mexico.

The critical problems faced by the treatment sector in Mexico, including the state of municipal wastewater reuse, was previously discussed by de Anda and Shear [14,15], de la Peña et al. [16], and by de Anda-Sánchez [5]. The critical problems in the municipal wastewater treatment programs and the new findings reported in this work are classified according to four perspectives based on an adapted Kaplan and Norton Balanced Score Card (BSC) model applied to measure the performances of the municipal services [17–20]. Later, the most relevant issues in each BSC area were identified. A figure was constructed to visualize the general strategy oriented to improve the provision of the treatment services in the country.

Three generally recognized dimensions of sustainable development have been well-enunciated: ecological, social, and economic [21,22]. In addition to these dimensions, Pawłowski [23] suggested including four more dimensions—namely, moral, legal, technical, and political. For the water and treatment projects, McConville and Mihelcic [24] proposed five dimensions that affect sustainable development: sociocultural respect, community participation, political cohesion, economic sustainability, and environmental sustainability. The authors explained that, in this context, the application of the five sustainability factors throughout the project life cycle will result in the design and implementation of appropriate technology. According to this, the sustainability dimensions proposed by Pawłowski [23] and McConville and Mihelcic [24] are strongly correlated. On the other hand, Saad et al. [25] acknowledged that the sociological factors, including community participation, public involvement, social perception, attitudes, gender roles, and public acceptance, would lead to improvements in wastewater management practices. To discuss the reported results, an analysis of six dimensions of sustainability was included. Finally, the progress in the country to adopt the Integrated Water Resource Management (IWRM) principles was considered [26].

## 3. Results

Table 1 shows the trend in population growth from 1950 to 2010 and the expected trend to 2050. The official data indicate that the country experienced a gradual decrease in the population growth rate from 2.7% in 1960 to 2.0% in 2010 [12]. According to the official population projections, it is estimated that the population growth will decrease by 2050 until it reaches 0.23%, and the total population will be around 150 million people. The equation used to estimate the average annual growth rate ($GR$) for every ten-year period ($n$), where $P_n$ and $P_{n-1}$ are the populations at the start and end of each decade, respectively, was:

$$GR = \frac{1}{10}\left(\frac{P_n - P_{n-1}}{P_{n-1}}\right)100 \tag{1}$$

**Table 1.** Historical growth of the Mexican population from 1950 to 2010 and the expected growth in 2050 [12].

| Year (n) | Total Mid-Year Population (P) | Average Annual Growth Rate (%) |
|---|---|---|
| 1950 | 27,026,573 | |
| 1960 | 36,786,543 | 3.61% |
| 1970 | 50,778,729 | 3.80% |
| 1980 | 67,561,216 | 3.31% |
| 1990 | 84,169,571 | 2.46% |
| 2000 | 98,785,275 | 1.74% |
| 2010 | 113,748,671 | 1.51% |
| 2020 | 127,792,286 | 1.23% |
| 2030 | 138,070,271 | 0.80% |
| 2040 | 144,940,511 | 0.50% |
| 2050 | 148,209,594 | 0.23% |

In Mexico, a population center is considered urban when it has more than 2500 inhabitants; where there are fewer than this, it is considered rural [4]. Figure 3 shows the historical evolution of the urban population in Mexico. At the beginning of the 1960s, the relationship between the urban and rural population in Mexico was close to 50% each. However, like global trends, Mexico has been urbanizing. In 2018, 80.2% of its citizens lived in urban areas. This figure is well above the world average of 55.3% in 2018 [27].

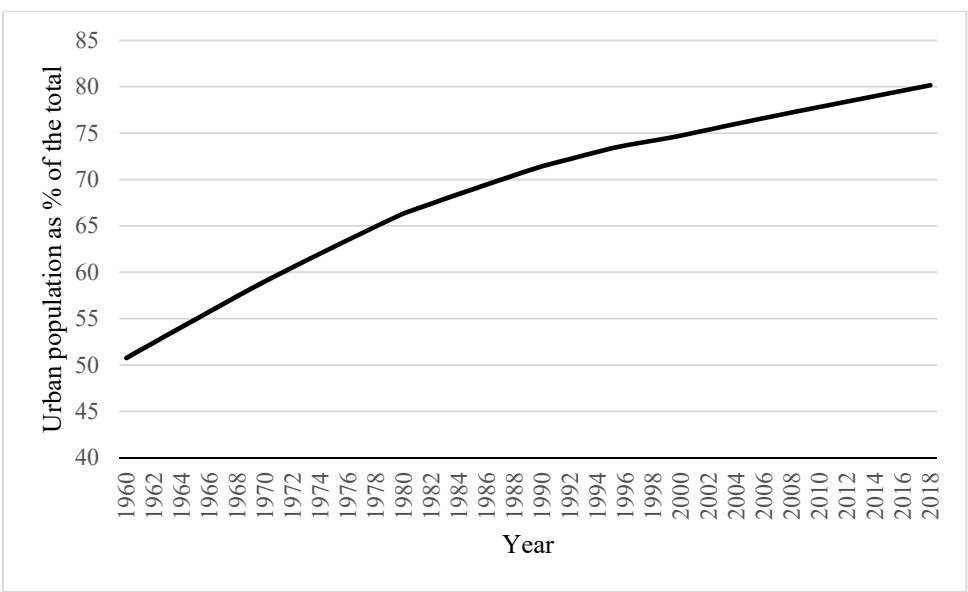

**Figure 3.** Historical growth of the urban population in Mexico from 1960 to 2018 [12].

### 3.1. Consumptive Water Use

'Granted' water in Mexico is defined as the concessional volume of water (underground and surface) for public supply. It corresponds to the volume authorized for use or exploitation of the water resource for public supply and domestic use. It includes the population that has a connection to a domestic, public distribution system, a protected well, or a rainwater collection network (Figure 4) [4]. In 2017, the volume granted by CONAGUA for consumptive use in Mexico was 87.84 hm$^3$. Approximately 60.9% of the consumptive water use in Mexico came from surface sources (rivers, streams, and lakes), while the remaining 39.1% came from aquifers [4]. The largest volume granted for consumptive use was for agriculture, mainly for irrigation, as shown in Table 2. It is important to note that, in Mexico, because of inefficient irrigation technology, about 60% of the water used in agriculture is lost through evaporation or subsoil infiltration. On the other hand, approximately 80% of the drinking water that is used by the population for domestic or commercial use turns into wastewater normally ending in the urban sewage network system [4].

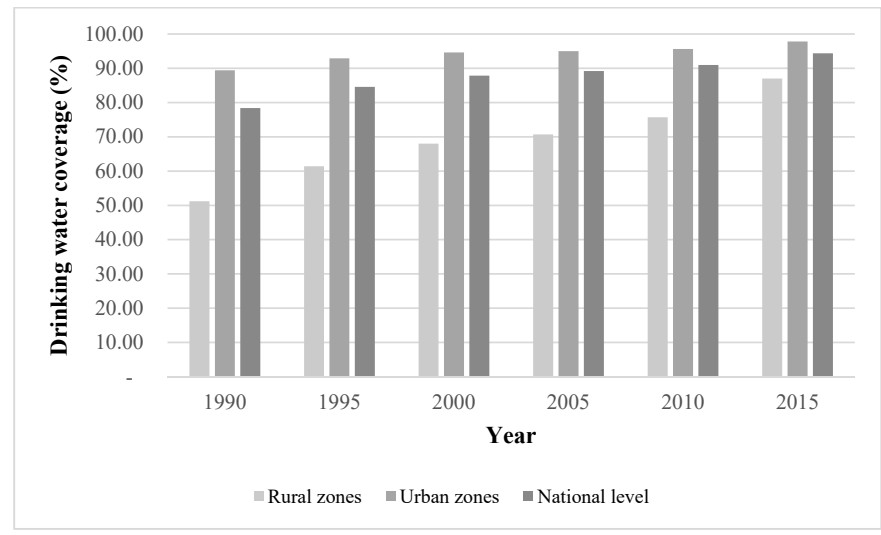

**Figure 4.** Evolution of the drinking water coverage in rural and urban areas in Mexico in the period from 1990 to 2015 [4,13].

**Table 2.** Consumptive uses grouped by source type for 2017 [4,13].

| Water Use | Origin | | Total Volume | Extraction Percentage |
| | Surface Water | Groundwater | | |
| | Thousands of hm$^3$ | Thousands of hm$^3$ | Thousands of hm$^3$ | % |
| --- | --- | --- | --- | --- |
| Agriculture | 42.47 | 24.32 | 66.80 | 76.04 |
| Public supply | 5.25 | 7.38 | 12.63 | 14.38 |
| Self-supplying industry | 2.04 | 2.23 | 4.27 | 4.86 |
| Electric power (without hydroelectric power) | 3.70 | 0.45 | 4.15 | 4.72 |
| Total | 53.46 | 34.38 | 87.85 | 100.00 |

### 3.2. Sewage and Treatment Coverage

In Mexico, the sewage coverage includes people who have a connection to the sewage network or to a septic tank; it does not necessarily include the conduct of wastewater to a treatment plant. According to CONAGUA [4], the evolution of sewage coverage in the country has been improved from 80% in 2000 to close to 92% in 2017. As shown in Figure 5, the trend has been almost asymptotic since 2012. The registered data on the number of municipal WWTPs in operation and out of operation in Mexico are described in Figure 6 for 1991–2017 [4,13]. An out-of-operation WWTP in Mexico includes any facility that the municipality has partially or totally closed because of a lack of financial resources to keep it operating. With the increase in population, the number of municipal WWTPs has increased by an average of about 7% per year. However, as shown in Figure 4, at the beginning of the 1990s, almost 32% of the total number of municipal WWTPs were out of operation. This trend reversed until 2009, when, once again, there was a steady increase in the plants out of operation, reaching 22.5% in 2016, the last year when the government published such records.

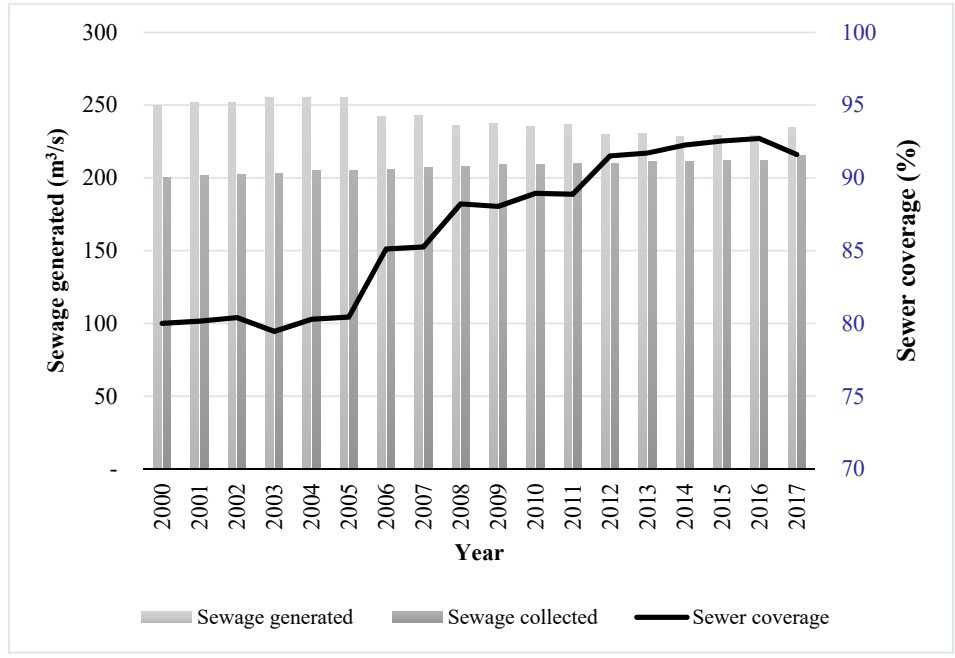

**Figure 5.** Historical trends of the sewer coverage in Mexico from 2000 to 2017 [4,13].

In Mexico, the installed capacity of a wastewater treatment plant means the design capacity, while the operating capacity is the record of the wastewaters received in the treatment system. In 2017, the total amount of treated municipal wastewater was 137,699 L/s (4342.42 hm$^3$/year). According to Figure 7, the percentage of the unused capacity of the municipal WWTPs has remained relatively low. In contrast, Figure 8 shows that there is still a significant gap between the wastewater generated in the country and the volume of treated wastewater. As can be seen in the same figure, the percentage of the treatment

coverage has been exponential, going from 17% in 1998 to almost 60% in 2017, the last year reported in the official figures.

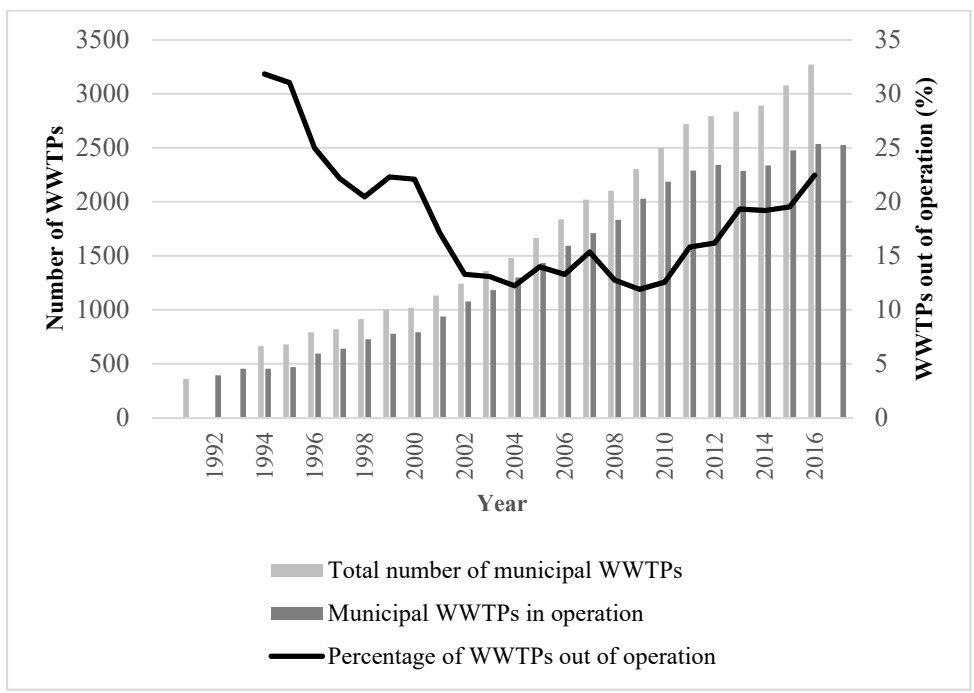

**Figure 6.** Historical trends of the municipal WWTPs in Mexico from 1991 to 2017 [4,13].

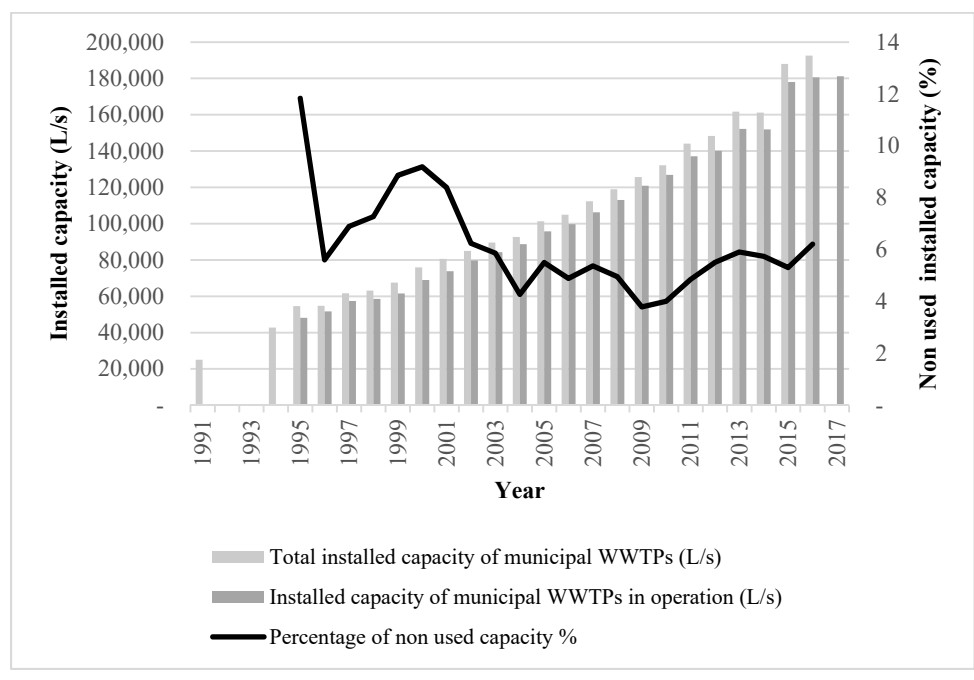

**Figure 7.** Installed capacity of the WWTPs in Mexico from 1991 to 2017 [4,13].

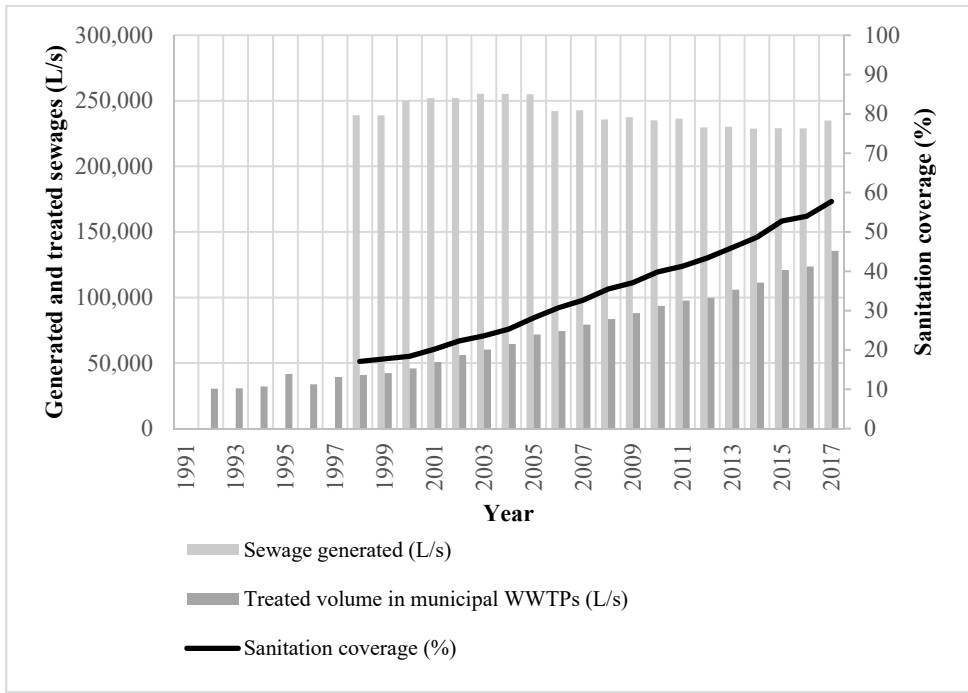

**Figure 8.** Gap between the wastewater generated and treated in municipal WWTPs, and the percentage of the treatment coverage from 1998 to 2017 [4,13].

### 3.3. Reuse of Treated Wastewaters

According to CONAGUA [4], in 2017, 39.8 m$^3$/s (1255.13 hm$^3$/year) of treated wastewater was reused directly from WWTPs and 78.8 m$^3$/s (2485.04 hm$^3$/year) was reused indirectly after first being discharged into a water body. Figure 9 explains the main paths of the water, wastewater, and reused treated wastewaters according to the Mexican water management model. As explained in Table 2, agriculture is by far the most demanding sector for water resources in the country, followed by public supply. The self-supplying industry and thermal electric power generation represent less than 10% of the entire water budget. In Figure 9, the dotted box for the water purification plants (WPP) indicates that, normally, in urban areas, there is the infrastructure to treat the water, but in rural areas, groundwater is used directly to supply the water needs of the population. The dotted box for the municipal and industrial WWTP means that, in some cases, the plant does not exist, exists but does not operate, or operates but does not meet the regulatory requirements. The dotted box in rainwater harvesting means that this infrastructure exists in some areas, since the collection of rainwater is not a priority in most urban municipalities in the country. The grey dotted lines represent diluted raw wastewaters or wastewaters after treatment, which are recycled for agriculture or the industry. The red lines represent the main sources for water pollution of the waterbodies in the country. Figure 9 shows how the discharge of raw wastewaters, municipal WWTPs, industry, energy production, and agriculture increase the pollution of water bodies in the country and how this polluted water is then used as a source of drinking water, thus increasing the public health risks and environmental pollution.

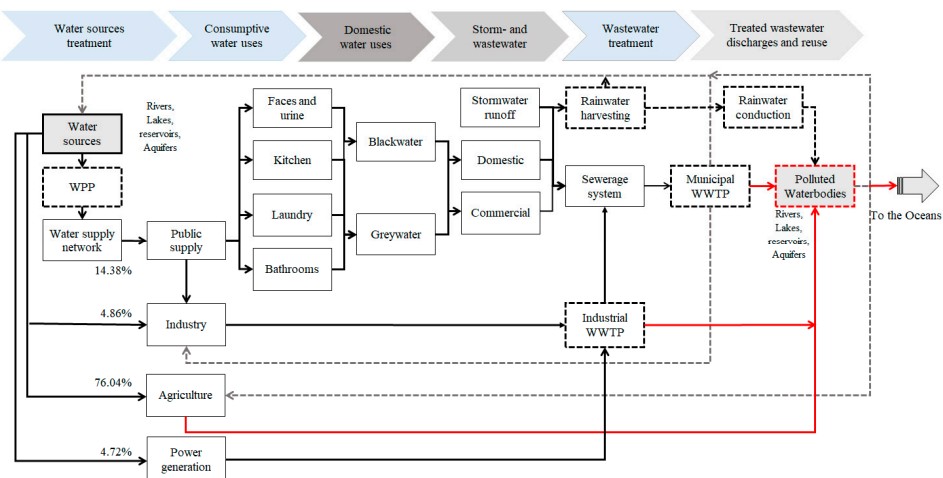

**Figure 9.** Main paths of the supplied water and wastewater in Mexico (Source: J. de Anda).

## 4. Discussion

The population projections for the next thirty years in Mexico estimate that the overall growth rate will decrease dramatically. The country must, however, cover the treatment needs of the wastewater generated by a growing urban population (Figure 1). According to a predictive forecast carried out by the National Population Council, the urban population in Mexico is going to increase by 2050 to 89.4% with reference to the population reported by 2018 (Figure 3) [12].

Even though sewerage system coverage has been developed significantly in the last twenty years, the treatment coverage of sewage and sanitation continues to be a challenge (Figures 5 and 8). In the last 20 years, the country has made a significant effort to increase the wastewater treatment coverage, going from just over 20% in the late 1990s to almost 60% in 2017, the last year officially reported. Wastewater treatment plants continue to grow in number; however, there are a significant number of facilities that do not operate (Figure 6). Figure 6 also shows that, at the end of 2016, in Mexico, the registry of municipal Wastewater Treatment Plants (WWTPs) showed 2536 facilities in operation, with an installed capacity of 180,569.72 L/s and a treated flow of 123,586.75 L/s, resulting in a national coverage in the treatment of municipal wastewater of about 54%. This figure is above the national average in other countries in Latin America, where most countries reported less than 40% coverage. However, a significant effort was made to reduce the number of facilities that were not in operation from 1990 to 2010. Unfortunately, this number has gradually increased since then, reaching 22.47% in 2016. This figure means that, in this year, there were 2536 treatment plants out of operation in the country.

In addition to these problems, one must note that, since 1996, the regulations regarding the discharge of sewage into national water bodies have not been modified. The official Mexican standard NOM-001-SEMARNAT-1996 does not regulate many of the compounds that today seriously compromise the health of the population and of surface and ground-water ecosystems such as organic micropollutants, pesticides and pesticide metabolites, urban and industrial organic micropollutants, veterinary compounds, and pharmaceuticals, among others [28]. Modifications to the current official regulations would put municipal treatment services in serious trouble because the technologies of the plants in operation are not capable of meeting even the more demanding requirements of these types of pollutants. That is why, to date, the official draft regulations have seen no progress for their approval in the Chamber of Deputies [29]. Consequently, at present, many waterbodies in Mexico exceed the limits of the most basic water quality parameters [4].

According to the previously mentioned methodology, the problems faced in the municipal wastewaters treatment in the country were organized based on the Balance Score Card (BSC) model. Figure 10 summarizes the results.

*4.1. Internal Process Perspective*

a. The urban concentration in Mexico today exerts a high to very high level of water stress and water scarcity in most cities located in Central and Northern Mexico [30].

b. There is still an important gap between the sewage generated by the population and the percentage of sewer coverage to collect this sewage [5].

c. Due to the lack of infrastructure for the management of urban rainwater, wastewater and rainwater are mixed in the same municipal sewage systems, meaning that opportunities for the aquifers to be recharged and for rainwater storage and reuse are lost [5].

d. Most municipal WWTPs are bypassed during the rainy season because they were not designed to handle the volumes associated with heavy rainfalls. As a result, during the rainy season, many of the receiving water bodies are contaminated by excess organic loads and other pollutants (personal communication with State Water Commission in Jalisco).

e. The leaks in the sewerage network are as high as 40% in urban areas [31].

f. More than 70% of the municipal wastewater treatment technologies installed in the country require intensive energy use and high Maintenance and Operational (M&O) costs, such as activated sludge, aerated lagoons, and dual anaerobic–aerobic systems [5].

g. There are limitations to the current regulations to control the nutrients and emerging contaminants [5,28].

*4.2. Users' Perspectives*

a. The interruption of drinking water services in cities during the dry season is more and more frequent.

b. There is increasing social discontent due to the high level of contamination of surface waters in Mexico and the resultant consequences for public health.

c. Farmers refuse to reduce their rights to water extraction volumes from their wells, because they could lose their government-approved water concessions if they agree to use treated water from the municipality.

d. Municipal politicians fear the loss of voter support if water metering is installed. At present, a single fee per user is applied, which is generally insufficient to pay for potable water treatment and for sewage treatment services. The consequence of accepting such a measure is the overuse and overexploitation of the local water resources and the contamination of surface- and groundwaters due to lack of, or inefficient, treatment infrastructures.

e. Water managers of municipalities with conventional WWTPs consider that the government incentives to cover part of their Maintenance and Operating (M&O) costs are insufficient to keep the facilities in good operating condition and that, in any case, the process of getting such incentives is highly bureaucratic.

*4.3. Financial Perspective*

a. The ground- and surface water volumes given in concession by CONAGUA for agriculture in Mexico cost nothing for farmers if they do not exceed the volume granted by CONAGUA, so, in most cases, farmers prefer to use well water than to use treated water from the municipality [32].

b. In some municipalities, the aquifers are overexploited, but the extraction of the water given in concession by CONAGUA to the farmers is still preferred, because, as noted above, there is no cost for the water, and the energy costs of pumping for agriculture are subsidized by the Federal Electricity Commission.

c. The centralization of the municipal wastewater treatment service involves the commitment to invest, build, and operate a complex facility requiring regular high expenditures for M&O.

d.    Medium-sized or large treatment infrastructure projects usually require development banking funds. With this, the municipality acquires long-term debts that, usually, it cannot pay.

e.    In most cases, the wastewater treatment plants do not have a wastewater reuse plan.

f.    Most municipalities do not have a decentralized water operating agency with financial independence from the municipality. As a result, the expected annual municipal budget for treatment is generally insufficient to maintain the WWTP M&O costs.

g.    Municipalities with a low population normally do not have water consumption meters and pay an annual fee for water services for housing independent of water consumption. This practice has led to the overuse of water, overexploitation of aquifers, and insufficient income to provide adequate potable water and treatment services to users.

h.    A lack of long-term political continuity in the municipalities to get financial resources with the state and federal agencies that manage the water resources affects the plans to expand the treatment coverage and the renewal of existing treatment infrastructures.

i.    Frequently, the managers of potable water and treatment services do not have enough information about the subsidies from CONAGUA to finance part of the M&O expenditures of the WWTP [33].

j.    Many municipalities do not have a treatment service for their wastewater and prefer to pay fines for unregulated discharges, because they do not want to incur financial debts or permanent M&O expenses.

k.    Knowing the financial limitations of the municipalities, CONAGUA usually defers the fines to municipalities that do not comply with the Mexican Federal Law of Rights declared in its article 276 [34].

*4.4. Learning and Growth Perspectives*

a.    Frequently, water managers do not have a basic educational background in subjects such as hydrology, laws, and the regulations related to the management of water resources or the basic principles of hydraulic urban infrastructure.

b.    Sporadic communication between municipal, state, and federal agencies results in the failure of water and wastewater management plans.

c.    In most cases, there is an absence of a vision of IWRM based on the hydrologic resources of the basins and subbasins and a lack of expertise and research programs to accelerate technology development and technology transfer activities that can offer innovative treatment technologies with low energy consumption and a low carbon footprint.

d.    Poor community participation in decision-making in the management of water resources results in limited confidence in the decisions of municipal representatives.

e.    A lack of transparency of the municipal authorities in the management of financial resources generates distrust in the community, and participation in community initiatives declines.

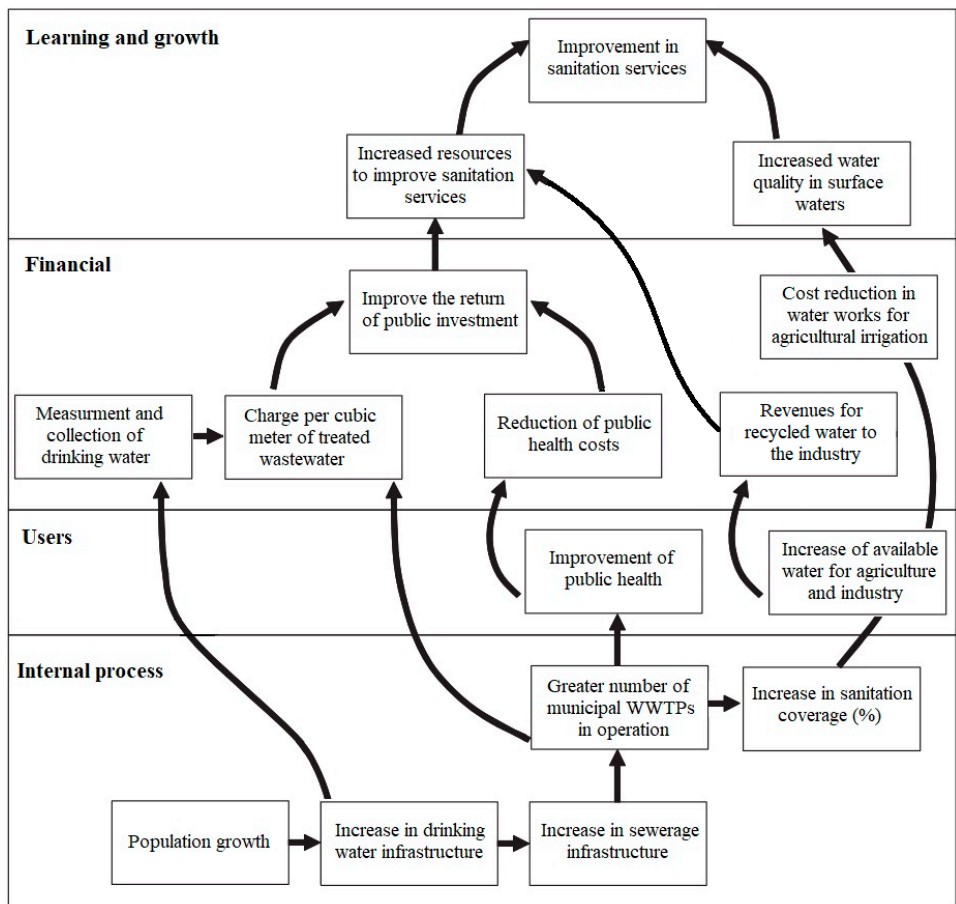

**Figure 10.** Balanced score card adapted to the sanitation sector (adapted from reference [35]).

### 4.5. The Sustainable Approach

The sustainable treatment and reuse of municipal wastewater in Mexico was analyzed in six dimensions: social, environmental, technological, economic, legal, and political [23–25,36].

#### 4.5.1. Social Dimension

The work published by Saad et al. [25] explained how to face the social issues related to municipal wastewater management. The publicity of new municipal wastewater-related projects includes advertisement in the media; education; and the inclusion of all stakeholders (politicians, experts, and the public at large). In the decision-making process, these are key elements for the successful design and implementation of wastewater schemes. Public involvement is best achieved through the participation and involvement of users in all parts of the project cycle, from planning and design to implementation and decision-making, thus producing more efficient and sustainable projects/outcomes.

On the other hand, the degree of acceptance of wastewater reuse varies widely, depending on the reuse purposes, and is influenced by many factors, such as the degree of contact, expressions of disgust, education, risk awareness, the degree of water scarcity or availability of alternative water sources, calculated costs and benefits, trust and knowledge, issues of choice, attitudes toward the environment, economic considerations, involvement in decision-making, the source of water to be recycled, and experience with treated wastewater. Other factors that depend on the region and case include the cultural, religious, educational, and/or socioeconomic factors [25]. On the other hand, the engagement of stakeholders in learning, planning, and acting requires a common sense of community, effective formal and informal collaborative processes, and the sharing of power and authority [37] (see Table 3).

**Table 3.** Pros, cons, and recommendations related to the social dimension in wastewater sanitation management.

| Social Dimension | | | |
|---|---|---|---|
| **Issue** | **Pros** | **Cons** | **Recommended Measures** |
| 1. Publicity | Essential to inform the public. Special interest groups may be aware of the plan, but not the public whose ongoing involvement is necessary for success. | Depending on the vehicle (radio, TV, print, etc.), this could be costly in terms of recruiting the public (see 2 below). | Develop a low cost but effective campaign to inform the public about the issues in any wastewater management plan. |
| 2. Public Involvement | Experience throughout the World (e.g., Great Lakes) has shown that public involvement is critical to the success of any sustainability program. | Sustaining public interest and enthusiasm over the long term is difficult. Who pays for travel and other expenses for the public participants? | Establish a Wastewater Management Committee (WMC), with funding, to facilitate public involvement. |
| 3. Degree of Acceptance | Essential for any long-term program. One needs the population to see that the program as necessary for them and for their children, grandchildren, etc. They need to accept the program as benefitting their economy, their health and wellbeing, their spiritual needs, etc. | Provision of costs/benefits may not be easy to translate into concepts that everyone in the population can understand. | The WMC will have the job of analyzing the costs benefits of using wastewater and translating this into language that anyone can understand. |
| 4. Actions of People | Achieving environmental improvement and sustainable communities will depend less on the mandates of government and more on the actions of people, communities, industries, nongovernmental organizations, landowners, and others, working together, often voluntarily. This will lead to a successful design and implementation of a Wastewater Management Plan. | To engage these stakeholders in planning, learning, and acting requires a common sense of community, an effective formal and informal collaborative process, and the sharing of power and authority. Ceding power and authority, when it has long been entrenched, will be difficult. | Experience has shown that when all sectors of society are involved in the development and implementation of a plan, power can be shared. Provide adequate funding for groups (farmers, small scale businesses, etc.) to participate as equals in a WMC. |
| 5. Role of Academia | Essential for the provision of scientific information to make sound decisions. | Academics may not see interacting with the public as part of their research career and may be reluctant to participate. | Once the WMC is established, have its members communicate with senior administrators in the academic community (government and university) to secure participation in the development of the wastewater plan from engineers, scientists, etc. |

### 4.5.2. Environmental and Technological Dimension

Wastewater treatment systems in developing countries like Mexico face several problems to keep them in operation. Most of these systems have been developed in countries with a high level of income per capita; high levels of technical expertise; and without considering the appropriateness of the technology for the culture, land, and climate of Mexico. Often, local engineers trained in the Mexican academic programs continue supporting the choice of conventional systems that later turn out to be inappropriate due to high M&O costs [38]. The successful employment of appropriate technologies requires a deep understanding of the social dynamics of the community in which they are applied [25].

In this sense, it is necessary to consider that the wastewater treatment processes are different in each case due to their different origins (e.g., effluents from agricultural and livestock activities and municipal or industrial discharges). In each case, the most convenient technologies must be clearly identified according to the results of the physical, chemical, and biological characterizations of the water to be treated [5,14]. Additionally, in the case of municipal wastewater, the selection of the appropriate technology to be implemented must be proposed according to the capacity of the community to pay for the M&O expenditures. Some measures to improve the environmental and technological dimensions of the problem are explained in Table 4.

**Table 4.** Pros, cons, and recommendations related to the environmental and technological dimensions in wastewater treatment management.

| Environmental and Technological Dimension | | | |
|---|---|---|---|
| **Issue** | **Pros** | **Cons** | **Recommended Measures** |
| 6. Combined domestic wastewater with rainwater | Reduction of urban sewage and rainwater infrastructure construction costs. | When domestic wastewater and rainwater are combined, they increase the demands for treatment capacity and, therefore, the costs of investment, maintenance, and operation [36]. | Introduce in municipal construction regulations the separation of domestic wastewater from rainwater. Gradually generate urban infrastructure to separate wastewater from rainwater. |
| 7. In many cases WWTP discharges do not meet environmental regulations | In the absence of complaints and an efficient surveillance system, the government assumes that it is fulfilling its obligations correctly. | Permanent pollution of surface and ground water and harmful effects on the health of people and the ecosystem. | Control the quality of the treated water so that it consistently meets environmental regulations [36]. |
| 8. Current wastewater treatment technologies produce important volumes of biological solids that require special treatment and disposal. | Most of the biological solids from WWTPs are managed and disposed in landfills or reused in agriculture. | When biological solids are not properly disposed, they produce offensive odors to closely settled communities, contaminate surface and groundwater and soils, and generate a large amount of greenhouse effect gases. | Ensure that the biological solids generated are treated appropriately for their use as fertilizer (compost), instead of their disposal in landfills that do not comply with environmental regulations [36]. |
| 9. Current WWT technologies are intensive in the use of energy and have high operation and maintenance costs. | Current technologies are widely used throughout the country. Waste- water treatment plant staff are familiar with these technologies. | Several municipal WWTPs are out of operation throughout the country because of high M&O costs. | It is necessary to introduce sustainable technologies based on natural processes and low M&O costs [39,40]. |

### 4.5.3. Economic and Legal Dimension

In the development of a sustainable scheme of integrated management of municipal water resources, including the management of sanitary wastewater, it is essential that municipalities adopt the scheme of a decentralized public organization that manages the economic resources destined to generate the water services. At present, the fee that users must pay for the supply of potable water and sewerage services, which includes the costs of treatment of the residual water, are integrated into a single fee that varies according to the municipality. Some measures taken in developing countries to improve the economic dimensions of the problem are explained in Table 5.

**Table 5.** Pros, cons, and recommendations related to the economic and legal dimensions in wastewater treatment management.

| Economical and Legal Dimension | | | |
|---|---|---|---|
| **Issue** | **Pros** | **Cons** | **Recommended Measures** |
| 10. Most of the technological solutions for municipal WWTP require high energy input and expensive chemical additives which can have a significant impact on the operation costs. | These costs can be offset by the benefits that the community obtains from having clean and safe wastewater discharged into receiving waters. | Municipalities have short administrations (3 years), and they often fail to see the environmental and social benefits of having safe surface and groundwater. | Complying with the condition that the balance of a mass and energy of raw materials, products, and byproducts results in an economic, social, and environmental benefit without compromising the current and future resources of the community [39]. |
| 11. In some cases, municipalities acquire debts to build WWTPs and subsequently cannot afford financial and operating expenses. | It is possible to compensate the financial expenses through the sale of the treated wastewater. | In general, sanitation projects are not planned to recover and reuse the treated wastewater and the value of the investment is thereby lost. | Evaluate the real investment capacity of the community and ensure the necessary resources for its maintenance, operation, and reuse [24]. |
| 12. In Mexico, the costs of energy have increased in recent years due to reduction of domestic oil reserves and lack of investment in renewable energies. | This situation should promote the use of sanitation technologies based on natural processes. | Not enough national technical capacities to promote sanitation technologies based on natural processes with low energy consumption and low M&O costs. | Consider that the energy, maintenance, and operation costs will increase over time, it will be necessary to build technical capacities in sustainable sanitation technologies [24]. |
| 13. Most treated wastewaters are discharged into rivers, lakes, and reservoirs, losing the opportunity to reuse it. | Efficiently treated wastewater protects the quality of surface and groundwater sources of freshwater. | Most of the municipal treated water is not reused in agriculture or industry to lack of incentives. | Rethink the law regarding water use and tax incentives to favor the public, industry, and farmers who reuse reclaimed water in their activities. Implement an education program to make farmers and the public aware of the benefits of reusing water. |
| 14. Sanitation coverage in the country is lower than 60%. It increases few every year. | The programs for the sanitation of wastewater in the country have been maintained in recent years, although not in accordance with the needs of the population. | There is a growing interest from private initiatives to collaborate with government in the cleanup processes of the country's hydrographic basins. | Introduce government programs such as tax incentives for the private sector to invest in the treatment infrastructure for public potable water and treatment to reduce pollution and promote a culture of care and reuse of treated water at all levels of society. |

### 4.5.4. Political Dimension

According to Mestre [41], the Mexican federal administration should reformulate the objectives and treatment strategies, rehabilitate and modernize the large number of installed municipal WWTPs, and accelerate the efforts to improve the water quality and treat 100% of the effluent. To remedy this problem, and to improve the quality of water, Mexico should undertake a major federal–state program to rehabilitate treatment plants and

make major investments to achieve universal coverage to treat wastewater. To implement this, some recommended measures are proposed in Table 6 [41].

**Table 6.** Pros, cons, and recommendations related to the political dimension in wastewater treatment management.

| Political Dimension | | | |
|---|---|---|---|
| Issue | Pros | Cons | Recommended Measures |
| 15. Most national waterbodies in the country face different contamination levels. | There is increasing awareness of the problem in the government, society, and academia. | The existing governance strategies have not been effective in increasing the sanitation coverage throughout the country. | Adjust the roles of federal, state, and municipal governments to improve the water quality. The states must assume the responsibility of reinforcing the governance schemes to accelerate the participation of society and academia in the decision-making process. |
| 16. Most municipalities pay very low or no taxes for sanitation. | People are happy not to pay higher taxes. | The population does not perceive in the short term the environmental damage and its impacts that are being generated by the lack of sanitation. | To convince the municipalities to charge the users appropriate fees for wastewater treatment services. |
| 17. Investment in sanitation services remains limited. | Public resources could be being applied to other higher-priority programs, such as insecurity and fighting poverty. | Gradual loss of water security in different regions of the country. | Make a significant effort to attract funding to invest in new WWTPs, including tertiary treatment. |
| 18. Lack of transparency in the use of public resources to keep sanitation infrastructure in operation. | This practice has allowed some politicians to use public resources for other programs without being held accountable. | The sanitation infrastructure is gradually being abandoned. | Make the origin and destination of subsidies transparent and accountable. |
| 19. Accountability at the municipal level often remains highly controversial and ineffective. | This has allowed politicians to make use of public resources to improve their political position. | The population's confidence in the government's ability to provide basic water supply and sanitation services is gradually being lost. | Create regulatory bodies at the state level and create a national coordinating entity, which has oversight and can regulate the subsector. |
| 20. With the change of municipal government every three years, the employees of the drinking water and sanitation services change. | A new work team enters in which the new municipal government has confidence in its performance. | Capacities created by the previous government are often lost. | Depoliticize the water management organizations so that the permanence of the managerial personnel is not connected to the renewal of municipal administrations. |

The strategy proposed by Mestre [41] to improve the wastewater treatment in Mexico includes measures focused on strengthening local government decisions and increasing the federal–state budget to subsidize the construction and operation of municipal WWTPs.

In the National Hydraulic Program 2014–2018, CONAGUA outlined the main lines of action to implement IWRM in the states; however, today the level of implementation of the program has reached a medium-low level, according to a UNEP [26] assessment. One

year after starting the new Federal administration, the new 2019–2024 National Hydraulic Program is still under discussion [42].

## 5. Conclusions

The main issues to solve in terms of wastewater treatment and reuse in Mexico should be focused on (a) increasing the coverage of wastewater treatment facilities and the sewerage system in urban areas and sewage treatment in rural areas; (b) improving the operational state of wastewater infrastructures; (c) planning the wastewater treatment system to suit the conditions of each municipality; (d) favor the reuse of treated wastewater in the Federal Water Law for agriculture, industry, and municipal services instead of groundwater use; (e) building capacities in wastewater management; (f) increasing the efforts to secure funds specifically for wastewater treatment; (g) increasing the capacities of overloaded treatment facilities; (h) ensuring the compliance of wastewater discharges with the required regulation for agricultural reuse; and (i) monitoring for compliance with the recommended guidelines.

The challenges that the sector must face in Mexico are not only in public funding and the environmental, technical, and organizational perspectives but also in the existing legal framework, lack of transparency, accountability, and public participation. Therefore, some additional measures to those mentioned above to improve the management of municipal wastewater and treatment are making transparency, participation, and accountability in budgeting and planning a reality at the national, state, and municipal levels. Reusing treated wastewater should be planned from the beginning of every project, and the energy and M&O requirements to facilitate decision-making should be considered in each project.

In terms of transparency, it is necessary to (a) publish the federal and state funds available for the wastewater treatment program, (b) to improve the transparency of procurement processes, (c) to improve the staff recruitment processes in the water and treatment sector, (d) to improve the management of municipal resources for the water and treatment sector, and (e) to report the annual environmental indicators related to water and treatment to the community.

To increase the sustainable use of water resources, it is also necessary to train municipal politicians and employees in the creation and maintenance of their own operating agencies and to strengthen governance models for the self-management of resources at the municipal level.

**Author Contributions:** Conceptualization, J.d.A. and H.S.; methodology, J.d.A. and H.S.; software, J.d.A.; validation, J.d.A. and H.S.; formal analysis, J.d.A. and H.S.; investigation, J.d.A.; resources, J.d.A.; data curation, J.d.A.; writing—original draft preparation, J.d.A.; writing—review and editing, H.S.; visualization, J.d.A.; supervision, H.S.; project administration, J.d.A.; funding acquisition, J.d.A. All authors have read and agreed to the published version of the manuscript.

**Funding:** Centro de Investigación y Asistencia en Tecnología y Diseño del Estado de Jalisco, A. C.

**Acknowledgments:** The authors acknowledge the Centro de Investigación y Asistencia en Tecnología y Diseño del Estado de Jalisco, A.C. and the University of Toronto–Mississauga for providing the time and information resources to write this work.

**Conflicts of Interest:** The authors declare that there are no conflict of interest in the present publication.

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
