# Peer review of "Sustainable Wastewater Management to Reduce Freshwater Contamination and Water Depletion in Mexico"

_water, doi:10.3390/w13162307_

Round 1

Reviewer 1 Report

In my opinion the article has potential, but the form in which it is presented means that the reader learns little about Author’s results obtained.

Comments related to the content of the article are the following:

  1. As Authors mentioned 'The purpose of present work is to discuss some of the reasons why...'. Unfortunately, the paper fails to undertake a discussion. The authors present numerical facts and generally known problems related to wastewater management in Mexico.
  2. Formula (1) lacks explanation of Pn and Pn-1
  3. Figures 3- 8 present historical data that are too far out to 2021. Supplementation with data from 2015/2017 -2020 would be required.
  4. Not understood is the sentence (lines 158-160) "...because of lack of irrigation technology it is estimated that about 60% of the water used in agriculture is lost through evaporation or subsoil in filtration and approximately 80% of the drinking water used by the population becomes into wastewater". Evapotranspiration and infiltration are natural processes that return water to the environment. What do the authors mean by 'lost'? How much did the plants use?
  5. The article lacks a 'Results' section. There is 'Materials and Methods' followed by 'Discussion'.
  6. In the 'discussion' chapter, the Authors refer in their analyses to archival data (2016; 2017) without considering the current state (at least 2020)
  7. It is difficult to find in the article thoughts, conclusions of the Authors, it is rather adopted conclusions of the situation from other publications or publicly available reports. This is also evidenced by the large number of citations when listing potential reasons for the current situation in Mexico

Author Response

  1. As Authors mentioned 'The purpose of present work is to discuss some of the reasons why...'. Unfortunately, the paper fails to undertake a discussion. The authors present numerical facts and generally known problems related to wastewater management in Mexico.

Disagree, we have an extensive discussion section where we analyze the problems and propose solutions.

  1. Formula (1) lacks explanation of Pn and Pn-1

Agree, we have added the explanation of the components of the formula.

  1. Figures 3- 8 present historical data that are too far out to 2021. Supplementation with data from 2015/2017 -2020 would be required.

Agree, but unfortunately since the new federal administration arrived, the National Water Commission (CONAGUA) suspended the publication of statistical reports on water in Mexico.

  1. Not understood is the sentence (lines 158-160) "...because of lack of irrigation technology it is estimated that about 60% of the water used in agriculture is lost through evaporation or subsoil in filtration and approximately 80% of the drinking water used by the population becomes into wastewater". Evapotranspiration and infiltration are natural processes that return water to the environment. What do the authors mean by 'lost'? How much did the plants use?

Agree, we have amended the wording in the text to explain better the mentioned figures.

  1. The article lacks a 'Results' section. There is 'Materials and Methods' followed by 'Discussion'.

Disagree, the results section starts on line 124 and extends until the line 222 of the amended version.

  1. In the 'discussion' chapter, the Authors refer in their analyses to archival data (2016; 2017) without considering the current state (at least 2020).

Agree, as explained on question 3, the information we use in the manuscript was the very last public available information. The official information after 2018 has not yet been published by the governmental agencies.

  1. It is difficult to find in the article thoughts, conclusions of the Authors, it is rather adopted conclusions of the situation from other publications or publicly available reports. This is also evidenced by the large number of citations when listing potential reasons for the current situation in Mexico.

Disagree: we do have conclusions of our own and we are recommending a clear path forward based on official data, on the oppinion of experts, and on previous published papers that have been analyzed and widely discussed in the manuscript.

Reviewer 2 Report

After a cerful study of the manuscripton: Sustainable Wastewater Maanagment to Reduced freshwater Contamination and Water Depletion in Mexico"

  1. The  manuscript does not bring anything new to science, but is a cry for help for people who lak clean water.
  2. References is missing position 42, which was cited at position 436.

Author Response

  1. The manuscript does not bring anything new to science but is a cry for help for people who lack clean water.

As explained in the manuscript, the purpose of present work is to discuss some of the reasons why, after years of investment in wastewater treatment projects in Mexico, no substantial progress in sanitation coverage has been made in recent years. We discuss the causes, consequences, and the measures that could be implemented to reduce the sanitation gap in Mexico. Learned lessons could be taken for other developing countries to avoid the current failures that the Mexican sanitation model has.

  1. References is missing position 42, which was cited at position 436.

Agree, we fail adding on publication on the reference [23], we add it and modify the number sequence of the reference section.

Reviewer 3 Report

The manuscript presents a problem, i.e. limited efficiency of wastewater management (recycling and reuse) system based on demographic projections in Mexico, while the solutions proposed are mainly socio-economic and political. The study seems well-documented and the article well-written.

Author Response

  1. The manuscript presents a problem, i.e. limited efficiency of wastewater management (recycling and reuse) system based on demographic projections in Mexico, while the solutions proposed are mainly socio-economic and political. The study seems well-documented and the article well-written.

Thanks, No comments.

Round 2

Reviewer 1 Report

In my opinion the article  has been redrafted, but the Authors still refer in their analyses to archival data (2016; 2017) without considering the current state (at least 2020). This remark is not involved in revised version of the paper.

Reviewer 2 Report

Dear authors, thank you for comprehensive answers to the questions contained in the review.